# The Collective Dimension in the Activity of Physical Education Student-Teachers to Cope with Emotionally Significant Situations

Magali Descoeudres 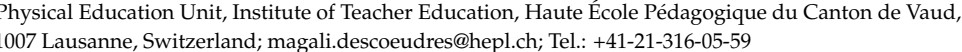

Physical Education Unit, Institute of Teacher Education, Haute École Pédagogique du Canton de Vaud, 1007 Lausanne, Switzerland; magali.descoeudres@hepl.ch; Tel.: +41-21-316-05-59

**Abstract:** The entry into the teaching profession is identified in the literature as a special, complex, and emotionally intense stage. Some teachers adopt turnover or attrition as coping tactics. The aim of this study is to understand the effect of the collective dimension on the professional development of physical education student-teachers in how they deal with emotionally significant situations. To avoid dropping out and to foster their well-being, beginners should develop their abilities by sharing their experiences. A mixed research design using a questionnaire and implementing a clinical activity procedure was adopted. Student-teachers (n = 139) had to write about 2 emotionally significant situations that they experienced during teaching, and they then shared or not with others. The second part of the study involved following up five student-teachers over the course of one year. A total of 32 filmed lessons with an emotionally significant situation served as support to self-confrontation and crossed interviews. The data were processed using the method of Bruno and Méard. The results show that student-teachers experience many emotionally significant situations, and this highlights the importance of using a collective dimension to help them cope with this emotional overload. Finally, these results open the potential value of a hybridised teacher education model of student-teachers, particularly when considering the emotional nature of the teaching profession and the necessity of sharing experiences in order to better deal with them.

**Keywords:** trainee teacher; physical education; interaction; emotion

## 1. Introduction

### 1.1. Novice Teachers between Turnover and Attrition

The entry into the teaching profession is identified in literature as a special, complex, and emotionally intense stage [1]. Some teachers adopt turnover [2] or attrition as coping tactics. Turnover means to change schools, while attrition insinuates leaving the profession [3]. Depending on the country, 10% to 50% of beginning teachers leave the teaching profession. In the United States, for example, 14% of novice teachers leave the profession after one year, 33% after 3 years, and 50% after only 5 years of teaching [4].

The emotional overload at the beginning of one's career [5] seems to generate an especially large drop-out phenomenon among novice teachers in Western school systems [6]. These early dropouts are linked to professional difficulties [7], emotional overload [1], and burn-out [8]. The image of a whirlpool [9] or a rollercoaster [3] can help illustrate this stage, which is marked by emotionally significant situations in the classroom experienced by teachers at the beginning of their careers [10]. While studies [11] have shown that children and students impact the emotional experiences of student-teachers, Lindqvist et al. [12] found it is possible to cope with the emotional overload by trying to modify the cause of the challenges. Moreover, other authors investigate how student-teachers' emotions are triggered by social interactions with their mentor teachers or their team student-partners [13]. Finally, attrition and turnover are seen as coping strategies [3] by some teachers, because

from the perspective of their identity construction, moments of feeling unsuited to being a teacher made them feel vulnerable [14]. Skaalvik and Skaalvik [15] show that teachers use five main strategies to cope with this job: hard working strategies, recovering strategies, reducing the workload strategies, job crafting strategies, and help-seeking strategies, mostly through interactions with others in the profession. Reports [16] of high attrition rates among novice teachers suggest that new teachers need help to develop coping strategies. This is best obtained while they are still trainee teachers under the supervision of a mentor. "Defining teaching as an ill-defined problem, where beginners have a limited repertoire of problem-solving strategies" [16] (p. 559), Le Maistre and Paré [16] suggest "that the ability to satisfice–that is, develop temporary but sufficient solutions–enables teachers to survive the early years of practice. However, it appears that, paradoxically, satisficing is one of the skills that is developed with experience" (p. 559). These authors show that expert teachers have learned how to cope with the experience. By mentoring, they could help beginners to deal with the complex problems generating intense emotions at the initial stages of their careers.

### 1.2. Interactions As a Way to Cope with Emotional Overload

Waber et al. [13] demonstrate the importance of emotions in interactive situations during school placement and the necessity of focusing on the emotional dimensions of becoming a teacher, in teacher education. They show, "that in different interaction situations, such as successful teaching-related cooperation, support, positive feedback, and goodwill of the mentor teacher, positive emotions are triggered, which are strongly connected to need fulfilment" (p. 520). On the opposite side, situations of failed communication, negative feedback, and lack of support generate negative emotions by the trainee teacher, as he feels a lack of guidance and the feeling of being poorly trained. Some recent findings [17] reveal that novice teachers experience conflicts with students, parents, and colleagues. To cope with these challenges, the new teachers adopt multiple strategies including collaboration with peers, colleagues, and tutors, as well as conformity, and autonomy.

The collective dimension through interactions in the teaching profession, especially by trainee teachers, can either be the origin or the consequence of novice teachers' emotions. To cope with the emotional overload and to prevent early professional drop-out, it seems that the collective dimension of teaching, when beginning teachers share their emotions with peers, colleagues, friends, or even the hierarchy, can help the novice at the beginning of their career [18]. Lindqvist [19] also confirms that, establishing relationships can be a way to cope with the multiple challenges novice teachers are confronted with during their first year as a teacher. Social support seems effective and is often illustrated as a pertinent solution to this problem. Moussay et al. [20] insist on the key role of the collective dimension at school in overcoming the dissonance between the ideal job and the real job. The gap between the ideal job and the reality experienced by these beginners produces several concerns, sometimes in opposition [21]. The student-teacher's actions generate intense emotions related to dilemmas, unpredictability, and the shock of reality [10]. In this study from Moussay et al. [20], we note that beyond the designated and official tutors, many other professionals (notably other colleagues who are not tutors) serve as interlocutors. These informal tutorial situations can also become a positive means for the novice to adapt or advance [20].

Another strategy includes having a critical friend [22]. Both possibilities of interlocutors are adequate to cope with teachers' emotions, either by being a critical friend or having one. The first strategy appears more fruitful [23]. This means that beginner teachers should themselves be a critical and supportive friend for another colleague, and not always only be the one needing to receive support. This could help create a more peaceful and easier transition into the teacher profession. It seems that much of the informational and emotional support is provided to trainee teachers from both their peers and teacher educators. Unfortunately, not all schools provide this integrated structure for social support [19]. The lack of social support is seen as *burdening* among trainee teachers [24].

### 1.3. The Case of Physical Education Teachers

Physical education (PE) in school curriculum supports student development of motor and social skills, and helps students make informed choices about lifelong physical activity. Physical education is a special subject taught at school, as Hébrard [25] said a long time ago. PE is a *discipline à part entière mais entièrement à part*, which means that it is a subject in its own right, but entirely separate from the other school subjects. PE is defined by Gaudreault et al. [26] as a 'marginal' school subject. One of the difficulties especially present in PE is the double management [27], of the entire class collectively and with students individually, as mainly related to the special needs of students [28]. Moreover, PE is taught in an open space without fixed and specific places for the students. While teachers of other subjects assign students to personal seats, PE teachers must interact with students in movement [29], which generates specific challenges, mostly because each student's involvement, successes, and failures, are visible to all. The literature identifies many concerns for PE teachers, for example: controlling the class [30], getting students to engage and avoid conflict [31], and adhering to the lesson plan [32]. These concerns create dilemmas providing intense emotions [33,34]. PE teachers seem to internalize unpleasant emotions in relation to the marginalization which could compromise their teaching [26]. However, [35] Salaveraa et al. emphasized that PE teachers have better control of unpleasant emotions because of their sporting background. They can find positive outcomes in negative situations through resource persons, tutors, peers, and colleagues [36]. In line with these specific resources and according to recent studies, PE teachers tend to have lower burnout scores and higher scores of engagement than other teachers [10].

Sharing with lecturers, tutors [37], or colleagues [33] seems to be a way to cope with emotionally significant situations experienced during school placement. Isolated teachers who do not share their experiences tend to leave teaching earlier [4], especially if they do not feel supported by their school manager. These results do not confirm the findings of [38] Jokikokko et al., who mention that, in cases of intense emotion, a novice teacher would adopt a silent strategy. However, other researchers emphasize the importance to novice teachers of exchanging experiences with colleagues in a supportive and professional context [13].

In this paper, we are going to discuss the collective dimensions of PE student-teachers' work in their daily practice. We will not discuss the collective teacher efficacy [39] and their measures, which are independent of the emotions felt as a novice and the linked potential for teaching professionals' dropout rates. We think that teachers' ordinary work consists of many collaborative aspects and that interactions with colleagues are key in coping with emotional overload and exhaustion, and for teachers' professional development.

### 1.4. A Clinical and Developmental Theoretical Framework

The clinical activity theory considers humans [40,41] from a developmental perspective and uses, among others, concepts from cultural historical psychology [42,43]. The authors postulate that the developmental process is based on the internalization of cultural signs, in the case of student-teachers, professional skills. In other words, our five student-teachers act according to developed motives (e.g., to manage students, to teach them) and mediation of tools (e.g., that help them to manage students, to teach them), received from previous generations [42]. The notion defended here is that these external cultural signs are first "learned" in a dissymmetrical situation (with a tutor, who is an expert at mastering this professional skill), then "grafted" gradually in increasingly symmetrical situations (with the help of colleagues), and finally mastered alone (without the tutor), thus becoming tools of thought and action for the development of one's own activity.

To make this second phase of development possible, this internalization implies psychic debates within the subject's consciousness, debates taking place between various simultaneous and competing options. Then, intrapsychic conflicts emerge because of exchanges and discussions between various persons, or they may develop because of unusual or unresolved situations that do not allow the subject to achieve the intended goals. These

initial intrapsychic conflicts often evolve into greater or more intense interpsychic conflicts that the subject repatriates into his own psyche. It is possible to imagine that these are also key elements contributing to the individual's development. In relation to our research object and our context, this theoretical framework allows us to estimate that PE trainee teachers receive "cultural signs" during discussions with their tutor, during exchanges with their colleagues, during their training, and/or during professional discussions with novice or experienced colleagues (the interpsychic). Then, they internalize these signs (the intrapsychic) to make them psychological tools that can be used in another context [44].

Considering emotions as the starting point of one's development, Clot [41] remind us that the individual's development is inseparable from their capacity to be affected by both negative emotions and positive ones, for example through perezhivanie, which means an unforgettable experience [45]. Our theoretical framework postulates that the power to be affected (to feel emotions) in each situation, is at the origin of potential development [46]. Thus, only longitudinal clinical research over a longer period, such as one year, permits us to analyze the developmental process inducted after experiencing emotionally significant situations. This clinical work contributes to a better understanding of the need and type of social support that teacher education must implement with the aim of helping novice teachers cope with the emotional overload and exhaustion present at the beginning of their careers [33], which may in turn reduce the phenomenon of professional dropout.

## 2. Materials and Methods

### 2.1. Study Design

A qualitative longitudinal research design based on a narrative questionnaire was adopted on one hand, and a clinical activity analysis was adopted on the other. The first method aims to understand with whom the emotionally significant situations [47] are shared, while the clinical study aims to investigate the professional development [48], of the student-teachers. In this paper, we will focus on one question from the first study and on the clinical research, with both linked to the collective dimension of the work.

### 2.2. The Aim of the Study

To better understand the developmental processes of PE student-teachers, we hypothesized that development takes place through *social* interaction [43] with many interlocutors. The research question consists of identifying the circumstances in which interactions with others fosters the development of PE student-teachers after experiencing emotionally significant situations.

### 2.3. Research Background and Participants

Both methods of internal research have been validated by the institution's Management Committee according to the Code of Ethics Research in the Swiss Universities of Teacher Education.

For the first study, a sample of 139 PE student-teachers (47 females and 92 males) across 3 flights, filled out the questionnaire during their first year of pedagogical specialization at the University of Teacher Education in Lausanne (Switzerland). The first flight included 45 students, the second flight 53, and a third flight contained 41. All volunteers, aged from 25 to 36 years old, had graduated from the University of Lausanne in Physical Education. No exclusion criteria were applied. During this pedagogical study, they learned to teach physical education in secondary schools (students from 10 to 16 years old) and in high schools/colleges (students from 16 to 19 years old). They also followed lectures at Universities for three and a half days every week. The rest of the time, they were in school placement and supervised by a mentor.

For the clinical study, 5 PE student-teachers from the county of Vaud in Switzerland (Anna, Carla, Luca, Luis, and Matt—the names of the student-teachers have been changed to provide anonymity—aged from 27 to 37 and part of the 139 student-teacher group from the first study) participated voluntarily in this research. These five student-teachers were

employed by a school during their placement, and they represent all the regions of the study context. The study design required that the five student-teachers filmed all PE lessons from one class for one year. If nothing emotionally significant occurred, the filmed lesson could be deleted. However, if an emotionally significant situation occurred, the participant was asked to contact the researcher for a self-confrontation interview (SCI) [41]. An emotionally significant situation was defined as a teaching situation that generated a positive or a negative emotion, and which the student-teacher reflected on afterwards. In this case, the participant had to keep the filmed sequence, to express in writing about the emotion and intensity felt, and to share this significant situation with others. According to the principles of the clinical activity [40], the data collection had several steps: (1.) The SCI interview took place less than one week after the emotionally significant situation occurred. (2.) Once each semester, cross-confrontation interviews (CCI) took place with two student-teachers and the researcher. (3.) A collective discussion with the five student-teachers and the researcher was then completed at the end of the school year, to close the research.

### 2.4. Data Collection

In the first study, 139 student-teachers reported, as was done in previous research [47], and 2 emotionally significant situations (n = 279) were experienced during their PE teaching while in school placement. These two situations could be classified as positive, negative, or transitioning from one to the other. The questionnaire took into consideration five main areas of significance: the valence, the emotion (Parrott's primary emotions) [49], its intensity, the sharing, and the effects from each situation. In this paper, we will focus on the sharing of experiences from this first study.

This study is based on *verbatim* transcription (made manually without software) from 24 SCI, 6 CCI, and a collective interview, spread over 1 whole school year. The interviews should allow the two researchers and the participants to assess at their real activity (*ibid*), i.e., the impeded part of the activity which was due mostly to dilemmas between what the student-teacher saw himself doing, what he would have liked to have done, or what he was unable to do. First, each participant was asked to contextualize the situation, describe it in detail, and then follow with a viewing of the filmed event. The description was then compared to the video sequence, both objectively and subjectively. Finally, hypotheses about the causes and opportunities of remediation were identified. The researcher first asked questions with "how" and then with "why", according to the collective dimension of trainee teachers, i.e., one of your colleagues observed this yesterday and you did it this way: what do you think about this other possibility? This fostered reflection, and new approaches were identified by the participants themselves because of their own learning from the video.

In our study, the SCI and CCI allowed participants to talk about conflicts, alternatives between several actions, as well as concerns related to their experiences as student-teachers. These intrapsychic conflicts could be the cause of them overcoming their initial approaches and shifting motives, which then generated activity development.

### 2.5. Data Analysis

The verbatim transcriptions coming from 24 SCI (total of 1582 speech turns), 6 CCI (total of 774 speech turns), and a collective interview, were processed according to the procedure from Bruno and Méard [48]. All participants completed one CCI except for Carla, who completed two because of the odd number of participants. Matt, Luca, and Luis undertook four SCI interviews, while Carla undertook three, and Anna completed nine. The data processing [48] procedure consisted of identifying the verbatim "development indicators" and coding them verbatim according to the eight concepts of clinical activity: auto-affectation [46], intrapsychic conflicts [41], varying between self-oriented or object-oriented activity [50], crossroads between different backgrounds [51], generalization [52], tension between the different instances of the profession [53], procedure between sense and efficiency process [20], and the creation of new goals [54]. This coding allowed the

researchers (double blind coding) to identify potential activity development by the student-teachers through the development indicators present following emotionally significant situations. Finally, the researchers confronted the few cases of different coding that required an additional one. In addition, the eight "big tent" criteria for excellent qualitative research were followed [55].

### 3. Results

*3.1. The Sharing of those Emotionally Significant Situations*

Almost all emotionally significant situations experienced by the student-teachers in school-placement were discussed with one or several persons (93.53%; 260/278).

The results presented in Figure 1 show that half of the situations discussed were shared with a tutor (151/278) or with colleagues (144/278), both being the preferred interlocutors selected by student-teachers to discuss an emotionally significant situation. Some are shared with the administrative hierarchy (31/278), and a certain number with non-professional interlocutors like a spouse (98/278) or friends (93/278). Unprofessional persons also offer opportunities to "share" an emotionally significant situation. The first study reveals that sharing with others is a way to cope with emotional situations experienced during the first year of school-placement, mostly with a tutor or colleagues, but also with non-professional interlocutors.

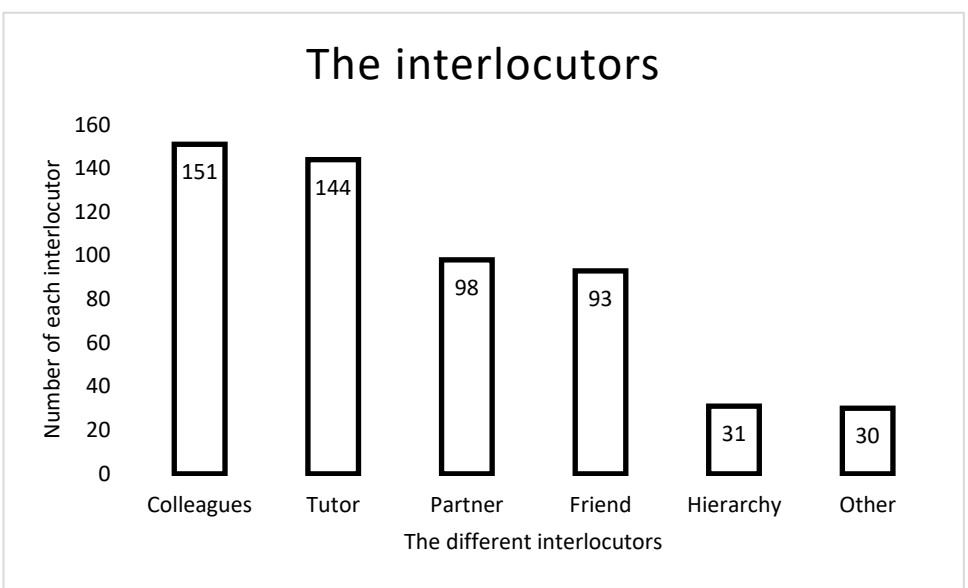

**Figure 1.** The interlocutors.

*3.2. The Interactions with Colleagues*

The results of the clinical study focus on the potential activity development of PE student-teachers in relation to their interactions following these emotionally significant situations. The verbatim show that two types of interactions have distinct effects on the development of PE student-teachers activity. We will first present the results relative to the interactions with their colleagues, and then present the results relative to their interactions with a tutor.

The five PE student-teacher participants of the longitudinal and clinical study contacted colleagues to help them process and deal with the emotionally significant situations experienced.

Anna: To cope with emotionally significant situations, Anna shares these situations with her colleagues because "it's important to share, so you can see that you're not the only one with this or that difficulty". Anna is comforted knowing she is not alone facing a

problem and moreover, she is helped by knowing that other professionals are also coping with a similar problem. The resulting impact is that she is auto-affected:

*"sometimes I think, I must ask to the referent teacher what he or she thinks about this student, because it's terrible! Then he or she says, don't worry, it's the mess in all lessons. It's helpful because I'm trying to find solutions and obviously, it doesn't work anywhere (laughs), so it's a little bit comforting."*

This sharing allows Anna to take a step back, to gain perspective, and to consider alternatives (development through efficiency with the construction of new operations) with the aim not to re-experience the same emotionally negative situations. It appears though that Anna does not really know how her colleagues are teaching, for example, how are they dealing with the students who are unable for many reasons to participate in a swimming lesson? She never talks with them about this specific problem because she feels intrapsychic conflicts, certainly because of the collective PE rule in her school that allows students to miss three lessons out of six. But Anna does not agree with this rule. Indeed, her disagreement is linked to the dangers presented when students are not able to swim after they leave their obligatory school years. Her concern and action are object-oriented instead of self-oriented.

Regarding the hierarchy, Anna feels a lack of support, even when she experienced negative emotional situations: "I wrote to the president of the school, because Tiago is not doing his sanctions. What else can I do? I see the class tomorrow and I'm still waiting for an answer".

The results show that sometimes the collective is a great support that permits Anna to relativize and to cope with emotionally significant situations. This sometimes hinders her actions, and she worries about these intrapsychic conflicts.

Matt: The interaction between Matt and his colleagues shows a particular mix of interactions. On one hand, he has interactions with his colleague with whom he is writing the essay (student studying in the same institute of teacher education), and on the other hand, he has his interactions with his colleagues from the school placement. Matt feels proud when his team partner observes him teaching a great PE lesson. As with Anna, Matt's interactions with colleagues foster or simultaneously hinder his activity development. For example, Matt's colleagues warn him against his intense involvement during his teaching: "if you want to last long-term without any turnover, you must be cooler". These colleagues collaborate with Matt but it is not mutual because Matt prepares the lessons plans individually for all the PE colleagues once a week, and teaches to all the students. Matt would like to collaborate more, but his colleagues are not interested, and this produces intrapsychic conflicts.

Luca: Luca regularly uses teamwork (PE colleagues and a reference teacher) when he shares a problematic situation, such as was experienced when one of his students used a badminton racket as a weapon. He also shares with his colleagues the lack of involvement from some students during a double Dutch assessment. Sharing enables Luca to confirm his opinions regarding theses emotionally significant situations. Luca often shares other experiences a well, such as the results from a cross-country event which enhances teamwork amongst colleagues. He shares experiences when it is also something pleasurable or even non-significant, and not just when he has had a difficult event. The collective dimension seems to be extremely positive for Luca.

Carla: Carla's experiences are yet also different, where sometimes interactions with her designated teacher support generate an emotional overload. Carla's teacher support should help with the two special needs students that she has, but this other person is not a professional teacher with PE either. She does not respond to the requests of Carla, even regarding the security at the trampoline jumps to secure the student with special needs. Instead of helping Carla, this co-teaching experience uses a lot of her energy and generates intense emotions. The resulting auto-effect hinders Carla's development. While the other PE student-teachers find help from colleagues to cope with emotionally significant

situations, Carla found her teacher support to be a source of her difficulty because of the lack of competency from them.

Luis shares with his colleagues and his tutor, the difficulties he is having in managing his special needs student who has autism. This student requires Luis' complete attention, and this disrupts his supervision of and teaching to the other students, thus fostering disruptive behaviors from them. This situation has caused Luis strong emotions and a feeling of loneliness. When Luis discusses this problem with the other student-teachers, they provide support and acknowledge that it is also difficult for them. This helps Luis to take a step back, and he says:

> *"Living such emotionally significant situations makes me doubt about the future, because I recognize that I'm unable to teach a student with autistic troubles while also teaching the whole class at the same time. That make me very uncomfortable during the lesson. I do not feel good, and I feel unprofessional. The only thing that helps me to survive is when I talk with my colleagues and they tell me that it's also difficult for them, even though they have more experience. It's a kind of relief for me. I'm almost happy to hear that I'm not the only one to experience these difficulties. My tutor gives me concrete recommendations that can help improve my lesson regarding this student with special needs."*

Our results show that there is a fine line between *attrition* or *turnover* and the continuation of teaching. Thanks to his colleagues and his tutor, Luis continues with his school placement.

*3.3. The Interactions with the Tutor*

Because the lessons were filmed, the five PE students-teachers could observe themselves not only with the researcher but also with their tutor.

During the SCI and CCI interviews, Anna watched herself teaching and expressed on many occasions, statements such as "I realize" or "I didn't think". Observing herself teaching causes Anna intense and unpleasant emotions: "When I watch myself teaching, I'm thinking, that's not possible (takes her head in her hands), my students don't move enough, like they are doing nothing and Tiago ... it's terrible (smiles while taking her head in her hands)". Anna asks her tutor if she has ever experienced a similar situation: "have you ever had a student who did that?" The interaction with the tutor in front of the video allows good development because Anna does not feel alone and can share situations that in the past her tutor also experienced. Anna implements new approaches, finds new alternatives, and often shifts or modifies her goals.

Our results show that Luis watches the lessons video with his colleagues and with his tutor. With the video, Luis understands better what he must improve upon based on the recommendations from his tutor. The guidelines from the institute of education recommend the tutor film the students at least once, so they can better understand the tutor's feedback and improve their teaching. Carla finds it difficult to observe herself on the video while she is teaching, and leaves an emotionally significant situation saying, "I almost hope that the video is no longer there (laughs) so that she doesn't end up in that embarrassing situation again". Carla feels real shame in how she taught the last PE lesson, "getting angry and shouting at students." By sharing the gym class management difficulties with her tutor, Carla can better cope with her strong emotions. Seeing themselves on the video creates a lot of emotions again, as if the student-teachers are re-living for a second time the emotionally significant situations. By sharing with their tutors though, they can see the benefits and identify alternatives with the aim of preventing or better handling similar and difficult experiences. From another perspective, when the situations experienced were positive, watching themselves on the videos allowed them to analyze the conditions and try to create these positive moments again.

Our results show that not all cases of student-teacher and tutor relationships are positive. Luca's interactions with his tutor are negative because his tutor gives very negative feedback and uses humiliation and moral destruction, which results in Luca considering stopping his school-placement. In the end, both Luca and his mentor find a

better way to communicate with each other. Overall, if the interactions between students and mentors are not fruitful, it is even more difficult to cope with the emotional overload from teaching. Sometimes, the emotional intensity comes from inappropriate interventions from the tutor, such as from non-constructive feedback, or when a student-teacher must teach dancing but has no direction or help from their tutor.

## 4. Discussion

The results highlight the importance of the interactional dimension in the development processes of PE student-teachers. In agreement with our theoretical framework, this activity's development takes place "in the social" [43] through various interlocutors, including colleagues and the tutor. Our results suggest that loneliness experienced during these training periods can lead to turn over or attrition [2]. The period at the beginning of one's career is a special, complex, and emotionally intense stage [1], often lived with authenticity [34]. This is difficult and can often be very lonely, but the moment of feeling unsuited to being a teacher makes them vulnerable, which is not bad for the student-teacher identity construction [14].

Regarding the PE context, readers are reminded that several simultaneous preoccupations coexist: respect for the lesson plan, a student with special needs or a "disruptive" student who requires greater attention and may be difficult to manage [28], and the ability to conduct simultaneous transactions with the whole class while also managing individualities in the gym, is a highly complex and variable operation that discriminates novice teachers from experienced teachers [27].

As some authors acknowledge, student-teachers often seek help from their peers [3]. In our study, participants seek help, resources, reassurance, or approval from their colleagues, especially regarding emotionally significant situations. In all cases, the student-teacher feels comforted that he or she is not alone in trying to cope with these difficulties. The group offers the student-teacher a type of echo that allows him/her to situate him/herself in the teaching community. Teaching gender [56] plays its usual role as a resource and as a safety net through exchanges with peers. Waber et al. [13] emphasize the impact of this collective dimension and of how the moments shared between trainers and trainee teachers fosters the development processes of their activities. The work of Moussay et al. [20] clearly shows that tutoring is always collective in school environments, and that it goes beyond the strict definition of an official tutor. According to these authors, colleagues can have a greater impact than the designated tutor, regarding the feelings of beginner teachers in being supported, as well as assisting on an operational level. In our study, this is the situation we observed among the five participants. In most cases, the sharing of each emotionally significant situation takes place with one or more colleagues who are not the designated tutors. The less instituted dimension of the relationship between the novice teacher and these "unofficial trainers" explains why the exchanges are more favorable, because they are less vertical. In concrete terms, when a tutor gives judgement or advice, this can be interpreted as another correction or directive, whereas advice given by peers does not have this prescriptive value.

Regarding the level of interactions with colleagues, Kayaoğlu et al. [57] emphasize the difficulty for the novice teacher of being confronted with other points of view, particularly in terms of the emotional charge relating to this type of exchange [57]. This was clearly the case for Matt in our results. Moreover, the literature supports our results that tensions can emanate from the relationships between experienced teachers and novice teachers [58], such as has been seen in a similar context in Japan. It is interesting to highlight that, in line with the literature, the novices in our study can describe the kind of teacher they would like to become. On the other hand, these novices often doubt themselves and turn to others for reassurance by asking colleagues if they already experienced similar difficulties that created significant emotions [59]. There is also a sense that the contradictions to this which were found in some of the literature are coming more from cultural differences (in Turkey,

Japan, France), which are based on implicit norms, unspoken authority relationships, and on teacher educations traditions.

Our results show that when emotionally significant situations are shared with others using this collective dimension, the student-teachers can better cope with these situations and better develop their teaching activities and approaches. Our results do not demonstrate the silent strategy [38], maybe because of our context. We note that the student-teachers' interactions with their students have an impact on the development of their activity. These impacts will be more favorable if they are shared with other interlocutors, notably colleagues.

## 5. Conclusions

The present study which was based on narrative questionnaires and a clinical activity analysis, aimed to explore the collective dimension of PE student-teachers sharing their experiences with others during their initial teacher training. More specifically, the purposes of the study were to: (1) identify if emotionally significant situations experienced by PE student-teachers are shared with others, and (2) identify in which circumstances the sharing of emotionally significant situations allows for professional development.

Concerning the first research issue, almost all emotionally significant situations experienced by the student-teachers in school-placement are discussed with one or several persons. Moreover, in connection with our second objective, the present findings show that the interactions with the tutor or colleagues permit the student-teachers to cope with the strong emotions.

This study has some limitations and opens new research perspectives. The collective dimension through interactions is only researched in the physical education field and could be extended in other disciplines. Thus, the number of participants in the clinical study could be increased.

Several perspectives can be mentioned to contribute to a longer and healthier involvement in the teaching profession. Firstly, the teacher education systems could be enhanced by these research methods to better develop the activities of novice teachers. Following the inputs about mentoring from Le Maistre and Paré [16], or about the critical friend aspect identified from MacPail et al. [22], some institutes for teacher education propose Certificates of Advanced Studies (CAS), with the aim to then become a tutor for beginning teachers. Support for teachers entering the profession is an important issue and is needed to encourage them to remain in the profession. Integrated into a cantonal system, this CAS provides a local and specific response to the support needs of the novice teachers.

In addition, to help novice teachers better cope with the strong emotions experienced during the lessons at the beginning of their careers, teacher education should introduce professional practice situations. A shared and collective analysis can be performed regularly after school-placement to ensure that these novice teachers do not feel alone. This will in turn foster greater professional development from all participants.

Finally, the complexity of the hybridized teacher training education system can be found in the collaborations and interactions of the participants (novice, tutor, mentor, lecturer) in the same project. A group activity analysis of emotionally significant situations [40,41] provides an opportunity to design hybridized teacher training education that supports work efficiency and the well-being of the student-teachers. These are essential factors in providing for a more stable and healthy continuation of the teaching profession.

**Funding:** This research received no external funding.

**Informed Consent Statement:** Informed consent was obtained from all subjects involved in the study.

**Data Availability Statement:** The data supporting the results, discussion and conclusions of this article are available at DOI: 10.13140/RG.2.2.36494.84800. Further inquiries can be directed to the corresponding author.

**Conflicts of Interest:** The authors declare no conflict of interest.

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
