# Peer review of "The Collective Dimension in the Activity of Physical Education Student-Teachers to Cope with Emotionally Significant Situations"

_education, doi:10.3390/educsci13050437_

Round 1

Reviewer 1 Report

Dear authors, some errors seen, or suggestions for improvement are shared:

Abstract

- It is convenient to add an introductory statement in the abstract, before the objective, to better justify it. Especially in reference to the desertion and turnover of teachers.

Key Words

- Avoid using words that are already in the title of the article. In this way, accessibility is increased and improved.

Introduction

- 96-98. Better define the subject of Physical Education. Not only the description of how it is, but what it is (its essence).

- 132-140. It is worth explaining this approach more clearly, especially for a reader who knows the field of study, but not that study approach.

- 156. It is suggested not to mention Spinoza since he is an antecedent of modern psychology, but his vision was totally metaphysical (specifically, in "Ethics demonstrated according to the geometric order"). It is suggested to use more recent scientific studies to support the statement.

method

- 168-171. It is suggested to support this methodological decision based on similar previous literature that has addressed this problem in a similar way, providing reasons.

- 180-209. The sample is explained. But it is necessary to describe and justify the sampling carried out.

- 219. How were the transcriptions actually made? How many researchers took part in the matter?

results

- Table 1 does not provide much information. By keeping it short and simple, it is better to express those figures in the text.

- Figure 1. It remains to add the title of the variable x and the variable y.

Discussion

- Future lines of research should be better indicated, as well as the limits of the present study.

conclusion

- An explicit conclusion section must be added.

Studies from the previous literature are relevant and up-to-date.

Congratulations are given to the authors for this study, which is considered relevant, and rigorous. But changes are necessary.

Author Response

The collective dimension in the activity of Physical Education Student-Teachers to cope with emotionally significant situations

Cover letter that explains, point by point, the details of the minor revisions requested to the manuscript and our responses to the referees’ comments.

First, we would like to thank the 3 reviewers that permitted to improve the quality of the manuscript with pertinent and coherent comments. Thank you very much.

Reviewer 1

Abstract

- It is convenient to add an introductory statement in the abstract, before the objective, to better justify it. Especially in reference to the desertion and turnover of teachers.

Thanks for your comments.

We added at the beginning of the abstract:

“The entry into the teaching profession is identified in literature as a special, complex, and emotionally intense stage [3]. Some teachers adopt turnover [4] or attrition as coping tactics.”

Key Words

- Avoid using words that are already in the title of the article. In this way, accessibility is increased and improved.

Thanks for the tip:

We took away: “emotionally significant situation”; “collective dimension” and added “emotion”

Introduction

- 96-98. Better define the subject of Physical Education. Not only the description of how it is, but what it is (its essence).

“Physical education in school curriculum helps students develop motor and social skills. And it helps students make informed choices about lifelong physical activity.” (lines 100-102)

- 132-140. It is worth explaining this approach more clearly, especially for a reader who knows the field of study, but not that study approach.

The clinical activity theory considers humans [1, 42] in a developmental perspective and uses among others, the concepts from cultural historical psychology [43, 44]. The authors postulate that the developmental process is based on the internalization of cultural signs, in the case of student-teachers, professional skills. In other words, our five student-teachers act according to developed motives (e.g., to manage students, to teach them) and mediation of tools (e.g., that help them to manage students, to teach them), received from previous generations [43]. The notion defended here is that these external cultural signs are first "learned" in a dissymmetrical situation (with a tutor, who is an expert mastering this professional skills), then "grafted" gradually in increasingly symmetrical situations, and finally mastered alone (without the tutor), thus becoming tools of thought and action for the development of one’s own activity.

- 156. It is suggested not to mention Spinoza since he is an antecedent of modern psychology, but his vision was totally metaphysical (specifically, in "Ethics demonstrated according to the geometric order"). It is suggested to use more recent scientific studies to support the statement.

Clot [42] refers to Spinoza (like Vygotski) to remind us that the individual’s development is inseparable from their capacity to be affected by both negative emotions and positive ones, for example through perezhivanie, which mean unforgettable experience (Fleer & Quinones, 2013).

Fleer, M., & Quinones, G. (2013). An assessment

perezhivanie: Building and assessment pedagogy

for, with and of early childhood science learning.

Dans D. Currigim, R. Gunstone, & A. Jones (Eds.),

Valuing assessment in science education:

Pedagogy, curriculum, policy (p. 231–247).

Dordrecht, the Netherlands: Springer.

method

- 168-171. It is suggested to support this methodological decision based on similar previous literature that has addressed this problem in a similar way, providing reasons.

The first method aims to understand with whom the emotionally significant situations [47] are shared, while the clinical study aims to investigate the professional development [2], of the student-teachers (lines 183-185).

- 180-209. The sample is explained. But it is necessary to describe and justify the sampling carried out.

We added further explanations. For the clinical study: These five student-teachers were employed by a school during their placement and they represent all the regions of the study context.

- 219. How were the transcriptions actually made? How many researchers took part in the matter?

This study is based on verbatim transcription (made manually without software) from 24 SCI, six CCI and a collective interview, spread over the one whole school year.

(…). The interviews should allow the two researchers and the participants to assess at their real activity (lines 236-238).

results

- Table 1 does not provide much information. By keeping it short and simple, it is better to express those figures in the text.

Yes, you are right. We removed it.

- Figure 1. It remains to add the title of the variable x and the variable y.

The titles of variable x and y have been added in Figure 1.

Discussion

- Future lines of research should be better indicated, as well as the limits of the present study.

The further lines of research have been clarified and the limits of the study presented (lines 497-516).

conclusion

- An explicit conclusion section must be added.

Studies from the previous literature are relevant and up-to-date.

We separated the discussion from the conclusion as the three reviewers asked.

Congratulations are given to the authors for this study, which is considered relevant, and rigorous. But changes are necessary.

Thanks for the pertinent comments which contribute to improve the manuscript.

Reviewer 2 Report

195 I propose to anonymize the names of the subjects!!!

311 I come back and say that the details are too personal and I recommend replacing names with initials or alter egos!!!

I propose a point 5, Conclusions, after 4 Discussions, which should clearly outline The aim of the study!

Author Response

The collective dimension in the activity of Physical Education Student-Teachers to cope with emotionally significant situations

Cover letter that explains, point by point, the details of the minor revisions requested to the manuscript and our responses to the referees’ comments.

First, we would like to thank the 3 reviewers that permitted to improve the quality of the manuscript with pertinent and coherent comments. Thank you very much.

Reviewer 2
195 I propose to anonymize the names of the subjects!!!

Page 5, there is a note saying: The names of the student-teachers have been changed to provide anonymity. Should we do more?

311 I come back and say that the details are too personal and I recommend replacing names with initials or alter egos!!!

As said, all names are alias names, and we didn’t mention from which schools we are talking about. We think it’s impossible to recognize either teachers or students

I propose a point 5, Conclusions, after 4 Discussions, which should clearly outline The aim of the study!

We separated the discussion from the conclusion as the three reviewers asked.

Reviewer 3 Report

The topic The collective dimension in the activity of Physical Education Student-Teachers to cope with emotionally significant situations is up-to-date, as it touches on the appropriate preparation for the implementation of physical education classes by young teachers. Young teachers encounter various problems at the beginning of their work at school. As a result, they often give up on it. This results, among others, from with the more and more frequently encountered educational problems of children and youth. Another problem is the lack of proper preparation for studies and support during work at school.

The subject of the article corresponds to the Special Issue "Current Challenges and New Perspectives on Physical Education", within the issue of PE teacher training.

The research methodology and results have been well described and do not raise any objections. It is only worth considering adding subchapter 5. Conclusions. The references to other research results described in Chapter 4 (Discussion) together with the presented conclusions resulting from own research should be separated.

I congratulate you on presenting interesting research that may be an inspiration for others to undertake the discussed research issues.

Author Response

The collective dimension in the activity of Physical Education Student-Teachers to cope with emotionally significant situations

Cover letter that explains, point by point, the details of the minor revisions requested to the manuscript and our responses to the referees’ comments.

First, we would like to thank the 3 reviewers that permitted to improve the quality of the manuscript with pertinent and coherent comments. Thank you very much.

Reviewer 3
The topic The collective dimension in the activity of Physical Education Student-Teachers to cope with emotionally significant situations is up-to-date, as it touches on the appropriate preparation for the implementation of physical education classes by young teachers. Young teachers encounter various problems at the beginning of their work at school. As a result, they often give up on it. This results, among others, from with the more and more frequently encountered educational problems of children and youth. Another problem is the lack of proper preparation for studies and support during work at school.

Thank you

The subject of the article corresponds to the Special Issue "Current Challenges and New Perspectives on Physical Education", within the issue of PE teacher training.

Thank you

The research methodology and results have been well described and do not raise any objections. It is only worth considering adding subchapter 5. Conclusions. The references to other research results described in Chapter 4 (Discussion) together with the presented conclusions resulting from own research should be separated.

Thank you

We separated the discussion from the conclusion as the three reviewers asked.

I congratulate you on presenting interesting research that may be an inspiration for others to undertake the discussed research issues.

Thanks for the pertinent comments which contribute to improve the manuscript.
